# Diagnosis of DSD in Children—Development of New Tools for a Structured Diagnostic and Information Management Program within the Empower-DSD Study

**DOI:** 10.3390/jcm11133859

**Published:** 2022-07-03

**Authors:** Katja Wechsung, Louise Marshall, Martina Jürgensen, Uta Neumann

**Affiliations:** 1Department for Pediatric Endocrinology and Diabetology, Center for Chronic Sick Children, Charité-Universitätsmedizin Berlin, Augustenburger Platz 1, 13353 Berlin, Germany; uta.neumann@charite.de; 2Division of Pediatric Endocrinology and Diabetes, Department of Pediatrics and Adolescent Medicine, University of Lübeck, Ratzeburger Allee 160, 23538 Luebeck, Germany; louise.marshall@uksh.de (L.M.); martina.juergensen@uni-luebeck.de (M.J.); 3Institute for Experimental Pediatric Endocrinology, Charité-Universitätsmedizin Berlin, Augustenburger Platz 1, 13353 Berlin, Germany

**Keywords:** DSD, information management, shared decision-making, genital variation, gender assignment, diagnostics, standardized care

## Abstract

Background: Current recommendations define a structured diagnostic process, transparent information, and psychosocial support by a specialized, multi-professional team as central in the care for children and adolescents with genital variations and a suspected difference of sex development (DSD). The active involvement of the child and their parents in shared decision-making should result in an individualized care plan. So far, this process has not been standardized. Methods: Within the Empower-DSD study, a team of professionals and representatives of patient advocacy groups developed a new diagnostic and information management program based on current recommendations and existing patient information. Results: The information management defines and standardizes generic care elements for the first weeks after a suspected DSD diagnosis. Three different tools were developed: a guideline for the specialized multiprofessional team, a personal health record and information kit for the child with DSD and their family, and a booklet for medical staff not specialized in DSD. Conclusions: The new information management offers guidance for patients and professionals during the first weeks after a DSD diagnosis is suspected. The developed tools’ evaluation will provide further insight into the diagnostic and information-sharing process as well as into all of the involved stakeholders’ needs.

## 1. Introduction

The acronym DSD (differences of sex development) refers to rare congenital conditions that affect the chromosomal, anatomical, or gonadal sex differentiation [1]. Genital variation is a clinical sign of some DSD diagnoses. It can prompt questions regarding gender assignment and treatment.

Historically, the “optimal gender policy” involved gender assignment to female or male and early surgery to “normalize” genital appearance [2]. Medical professionals made decisions and disclosed little to no information to the affected individuals (paternalistic decision-making). Individuals with DSD have challenged this approach, which profoundly changed the recommendations for care since the beginning of the 21st century [1,3]. Research has highlighted the requirements of children and families for a transparent communication and complete information about the condition [4,5]. Increasingly, genetic mutations are found in the diagnostic process. An exact diagnosis can be made. Evidence for the long-term outcome of medical interventions remains scarce for most DSD conditions. Decisions have to be made on an individual level [6].

Current guidelines therefore propose an informed shared decision-making approach with the active involvement of children and their parents [1,7,8].

Over the course of the shared decision-making process, health care providers and patients collaborate on the way to reach health decisions [9]. This process acknowledges that decisions against the background of uncertain evidence are influenced by the family’s emotional, social, and cultural background—likewise also by the health provider’s preferences [10]. Important elements of the shared-decision process are the provision of knowledge for the family and the joint discussion of risks and benefits of treatment options [11]. This type of decision-making aims to minimize decisional regret, as well as raise adherence to care and resilience [12].

Thus, education and information transfer are crucial in the care for children with DSD. The DSD-life study has identified information management as a priority of patients to improve care [13]. Information management in the DSD context comprises the information of patients and families by professionals, the information of children by their parents but also the information of the family’s social network. A need for “practical resources for patient/caregiver education and decision-making” was also noticed in a study that defined the principles of quality care for pediatric patients with DSD [14]. So far, no standardized information tools exist for children with DSD and their families. An information program that could assist professionals in the condition-specific education of children and their family is lacking.

The first weeks after a suspected DSD diagnosis are critical for children or adolescents and their families. Parents or older children rapidly seek information on the internet, which can be confusing and of poor quality [15,16]. Worries about the child’s identity, future perspectives and a wish for a rapid diagnosis can infer a sense of urgency and anxiety [17]. The affected children and parents face a number of challenges to understand the complex aspects of DSD and to find specialized medical as well as psychosocial care that can guide them in decision-making [18].

In Germany, several specialized DSD centers offer a patient/family-centered care by a multidisciplinary team, according to the current recommendations [7,8]. The interdisciplinary team organizes diagnostic procedures, psychosocial support, and transparent information concerning the child and their parents. So far, this information and care process for children and youth with newly diagnosed DSD, which might be associated with genital variation or questions of gender assignment, has not been standardized in Germany.

Within the study Empower-DSD, a structured diagnostic and information management was developed to standardize the care of children during the first weeks after a DSD diagnosis is suspected. On the one hand, the emphasis lies on the provision of existing knowledge on medical, social, legal aspects, and treatment options. On the other hand, the importance of psychosocial care and peer support is taken into account.

The present publication describes the development of the information management and presents the resulting structure, materials, and tools.

The information management addresses children of different age groups (newborn to adolescent child) and their caregivers. The term “family” is used to take into account the various constellations.

## 2. Methods

### 2.1. Study Design

Empower-DSD is a prospective longitudinal, mixed methods, non-controlled multicenter study. The study is funded by the German Innovation Fund of health insurance companies.

One objective of the study is to develop and evaluate an information management for children with a newly diagnosed DSD, which is associated with genital ambiguity or questions of gender assignment and their parents. The aim of information management is to foster the diagnosis-specific knowledge and empowerment of patients and their parents, thereby promoting shared decision-making. The information management purports to improve openness and coping with the diagnosis. Thus, the goal is to ensure the affected children’s optimal development and enhance patient satisfaction.

The other objective of the Empower DSD study is to develop and evaluate an age-specific group education program for children and adolescents with DSD and their parents [19].

### 2.2. Study Group

The Empower-DSD study group consists of 14 professionals (psychosocial and medical staff) working in specialized DSD departments of five university hospitals in Germany and representatives of German patient support groups for CAH and XX-/XY-DSD.

The Institute of Social Medicine, Epidemiology and Health Economics (Charité-Universitätsmedizin) is participating in the study for the qualitative evaluation. The Institute of Clinical Epidemiology and Biometry (University of Würzburg) is in charge of the central data management.

### 2.3. Development of the Information Management

A working group of pediatric endocrinologists and psychosocial professionals from two German DSD centers first developed a list of diagnostic procedures, medical, and psychosocial topics relevant in the first weeks after a DSD diagnosis is suspected.

The two centers’ experiences and structures were compared, common elements of care and topics for the information management identified. Initially, two target groups for gathering information (families and professionals) were determined. The group decided on the development of paper-based materials for these two groups. The elements of care concerning the first weeks were arranged in a timeline, the topic list grouped in this order.

The material for professionals was split into two separate items, when it became obvious that the need for information for professionals less experienced in DSD and members of the specialized team differed. The working group therefore decided on a structure in three separate, paper-based materials aiming for the different target groups: families, professionals outside a DSD center, and professionals in a DSD center. Drafts of these documents were prepared and structured according to the initial topic list from October 2019 to March 2020 in the working group. Finally, the material was circulated in the Empower-DSD study group. Further comments and suggestions were discussed and included if relevant. The Empower-DSD study group agreed on a common terminology and the use of an inclusive language, which was applied to all materials.

A consensus meeting with representatives of all study centers (at least one team member with medical and one member with psychosocial background) and patient advocacy groups was held in June 2020 to discuss open questions, organization issues, and, finally, to approve the information management´s structure and content.

### 2.4. Compilation of Evidence and Existing Materials

A PubMed^®^ search was performed for full-text English language articles using the search terms DSD/diagnostic guideline, DSD/standard, DSD/consensus, DSD/diagnosis, DSD/psychology, DSD/imaging, and DSD/genetics. Guidelines, consensus statements, and reviews published from 2010 to 2020 as well as relevant articles from the reference lists of those were selected.

The information management includes existing patient information. To obtain an overview of existing materials, an open-hand and online search were conducted. First, existing print patient brochures for families with newly diagnosed DSD used in the DSD centers were reviewed. Second, available print material (brochures and information leaflets that are downloadable as pdf) was searched online in January 2020 and again in March 2022 via Google using the search terms “DSD”, “Varianten der Geschlechtsentwicklung” (“variants of sexual development”), “intersex”, “AGS”, and “CAH”. The first 100 search results were considered. Materials were integrated if they were written in German, contained medical and/or psychosocial information relevant for the first weeks after a suspected DSD diagnosis with genital variation or questions of gender assignment and fulfilled the EQIP criteria for quality [20]. Moreover, the working group screened material in English for additional topics not yet incorporated in German brochures.

### 2.5. Evaluation of the Information Management

The developed information management will be evaluated. Children with suspected XX-/XY-DSD as well as sex chromosomal DSD and their surrogates with first appointments in four participating DSD clinics (Berlin, Lübeck, Bochum, Ulm) will be offered participation in the evaluation. The study aims for the inclusion of 30 children/adolescents and their parents.

For the quantitative evaluation, the diagnosis, the included patients’ age, and the information management process’s documentation will be analyzed with descriptive statistics. A qualitative evaluation with semi-structured interviews of the patients, their families, professionals, and peers will assess the expectations as well as hopes for the information management and how the participation was experienced. The evaluation will be performed after all included patients have completed the information management program. There is no interim analysis planned.

## 3. Results

The Empower-DSD information management organizes the diagnostic and information process for primary and specialized care after a suspected DSD diagnosis.

The materials’ medical and psychosocial information is based on national and international consensus statements, published expert opinions, and results of DSD research. In a literature search via PubMed^®^, 26 articles were retrieved. Twelve guidelines and general care recommendations were identified [7,8,13,20,21,22,23,24,25,26,27,28]. Moreover, articles with a special focus on the somatic, biochemical, and genetic assessment, as well as on imaging and psychosocial care in children with a suspected DSD diagnosis, were selected [29,30,31,32,33,34,35,36,37,38,39,40,41,42].

In the search for existing patient information, seven materials directed to parents and children were found. Four materials were selected to be passed on to the families during the information management (Appendix A). All included brochures covered topics related to a diagnosis in the newborn period. No information directed toward older children and their parents was found. There was only one booklet addressing health care professionals, i.e., obstetricians. It contained specific guidance for the communication with parents of newborn children. No material including communication skills with older children for primary care providers was identified. During the study, the web page https://dsdcare.de (accessed on 16 May 2022) was launched providing insightful information about existing materials for families in Germany.

Within the Empower-DSD study, three new paper-based tools were developed, addressing the needs of the different target groups that are involved in the diagnosis and care process:

One tool was created for staff not specialized in DSD. It provides general information about the first contact and the time until referral to a specialized DSD center. In addition, it includes a list of existing patient brochures. Two tools assist the care in the specialized DSD center: “My record”, a personal health record and information kit for the child/youth with DSD and their family; a guideline for the specialized multidisciplinary team containing templates with a checklist that aid the standardization as well as documentation of the information transfer to the family. Figure 1 illustrates the defined elements of care and information management tools. The following overview will outline the care process defined by the developed materials. It will introduce the new tools and their use in more detail.

### 3.1. A booklet for the First Contact in Primary Care

The suspicion of a DSD diagnosis mostly arises in a primary care setting. Genital variation at birth, absent pubertal development, or the incidental finding of abdominal gonads during surgery are examples where professionals like pediatricians, neonatologists, gynecologists, urologists, pediatric surgeons, midwifes, and nurses could suspect a DSD diagnosis in a child or adolescent. To address health care professionals with less experience in the treatment of DSD, a short 18-page booklet (written in German) was developed. The booklet is meant to provide an overview on important topics for this early period. The document aims to facilitate a calm environment for the family, minimize anxiety, and ensure rapid access to specialized care. In addition, it reassures professionals that the referral to a specialized center is an important step in the care process for children with these rare diagnoses. It advises the avoidance of unnecessary medical procedures. The booklet proposes an open, adequate communication with the families, and gives examples of useful phrases. It refers to existing brochures for parents with newborn children (Appendix A).

Table 1 shows a summary of the main topics that are included in the booklet. Information regarding the latter was distributed via the German national societies for pediatrics, gynecology, neonatology, and midwives. It is accessible via the Empower-DSD webpage (https://empower-dsd.charite.de, accessed on 16 May 2022).

### 3.2. Guideline and Checklist for the Specialized DSD Center

The guideline for the specialized multidisciplinary team defines structural requirements for the information management in the specialized DSD center. The multidisciplinary team should encompass pediatric endocrinology, psychology, pediatric radiology, and surgical subspecialty (pediatric surgery and/or pediatric gynecology and/or pediatric urology). Other disciplines can be involved as needed (e.g., social work, genetics, ethics, pediatric neurology).

Concerning each care element, the guideline specifies actions for the team and topics that should be discussed with the families. A checklist condenses all elements and corresponding topics (Table 2).

Each clinic’s multidisciplinary team complies with the structural requisites specified in the guideline and documents all consultations as well as discussed topics in the checklist with the current date.

The results of all assessments are shared with the family. To understand the health implications of the DSD diagnosis, the family is educated about sexual development and medical aspects. The child is involved in the information exchange in an age-appropriate manner. The team further addresses psychosocial as well as legal and administrative issues. In consultations with a surgical subspecialty, the risks and benefits of surgical procedures are discussed, if applicable. The family is educated about the options of surgical decision-making according to the change in German legislation from 2021. Surgical interventions on genitals must be deferred until the child can give informed consent, unless the surgery is life-saving. If a surgery is aimed to prevent harm to the child´s health or in case of a sinus urogenitalis in girls with CAH, approval for an early surgery must be requested at the family court.

The guideline defines a duration of eight to twelve weeks after the referral to the DSD clinic for the information management. All relevant topics of the checklist should be discussed at least once, preferably as often as the family requires within this period. However, the checklist does not specify the number of appointments or the sequence of care elements. The team should schedule appointments according to the family’s needs. Depending on the location of the DSD service in a rural or urban area and the distance to the family’s residence, appointments are made more or less frequently in each center.

A case manager, who is the primary contact for the family organizes the appointments and diagnostic procedures. The case manager´s professional background varies between the DSD centers of the Empower-DSD study group (e.g., pediatric endocrinology, psychology or nurse practitioner). The guideline for the specialized team defines the topics, which have to be discussed with the family. A medical expert covers medical topics and a team member with psychosocial background discusses psychosocial topics with the family.

### 3.3. My Record

In the specialized center, the case manager introduces the team members and explains the information management’s goals and process to the family. The team addresses the family’s questions, explains results of earlier investigations, if applicable, and outlines the planned examinations as well as diagnostic procedures. In due course, the team explains the challenges of decision-making in DSD, the purpose of psychosocial care, and potential peer counseling opportunities. If desired, a consultation appointment or meeting with a peer is organized. The team members describe the importance of the diagnostic and decision-making process documentation. The folder named “My record” is handed to the family. The sections are explained (Table 3). “My record” is a paper-based resource kit that combines a personal health record with information on key age-related topics in DSD. This tool is directed at the family (i.e., parents of a younger child) or older child/young adult. Therefore, it is written in a non-technical language. It provides guidance during the information management and beyond for life-long care. The folder can be adapted individually. If updates arise, the team and the family can easily rearrange or replace the paper prints within the folder or add further information.

The folder purports to favor medical information collection. Therefore, there are sections in which to file the results of medical assessments. The documentation in “My record” aims to assist the open discussion of all results and the transition of adolescents to adult care in the further course.

One section provides information on the proceedings during visits in the clinic and a template to take notes. This section’s covering page contains a short explanation on possible medical assessments and prepares the family for the next visits.

The information given in “My record” is structured chronologically, addressing questions that might arise during the newborn period (e.g., naming/sex assignment), extending to topics in kindergarten and later during the school career, puberty, and adolescence. This information timeline gives parents a short overview on medical and psychosocial issues in DSD that are important in the present or may be more relevant when the child is older (“ *How can I support my child regarding psychosexual development, sexuality and relationships, fertility, gender and society, social network, kindergarten, school, bullying, hobbies, personal strengths, and legal aspects?*”). Families are informed that results of investigations can remain pending for a long time. They are also educated regarding the fact that a genetic diagnosis may not always be made. The chapter explains that medical intervention and non-intervention are equally important in the care of children with DSD.

Parents can use the written information to formulate questions for the team and comprehend the concept of a long-term care process. Existing brochures developed by patient support groups and contact details for peer counselling are filed in a separate section.

The last section, “thoughts and feelings”, provides a space for parents of newborn or small children to collect memories about family values, information, and personal thoughts that played a role in their decision-making. This documentation can assist the passing of information from parents to children in an age-appropriate manner. Adolescents are encouraged to keep a diary for thoughts regarding their condition.

### 3.4. Templates for the Diagnostic Process and Team Meetings

To harmonize the diagnostic process in all study centers, the guidelines for the multi-professional team contain six templates, which can further aid the results’ documentation and discussion with the family. The templates summarize recent recommendations on medical history taking, physical examination, and biochemical and genetic assessments, as well as imaging. A photo documentation during the somatic assessment is not required and only possible after the family has given informed consent. Where photographs are stored must be explicitly discussed with the family. Ideally, a copy is placed in “My record”.

A template for the clinic visit’s documentation can help to retain an overview about who was present, which examinations were performed, and what topics were discussed. This template is included in the guideline for the team as well as in “My record” for the family.

A meeting of the multi-professional team in the specialized center is mandatory in order to discuss the assessments´ results for each patient and to develop a management plan. Furthermore, a case conference with a medical and psychosocial representative of each study center takes place every three months during the study. In these meetings, the team exchanges and collects information on assessments and explores the evidence for recommendations. A template that lists important topics for the team discussion was developed. Results of the team meeting and case conference are discussed with the family. Information on the proceeding of team meetings and the case conference are also included in “My record” for the family.

### 3.5. Exchange of Information between Primary and Specialized Care

The guideline proposes a feedback concerning the diagnostic procedures’ results to the referring and further treating primary care physicians. This report also includes recommendations for further care and treatment. In many cases, joint care is provided, in which families present to the DSD center at specific intervals but are primarily seen by a family physician, pediatrician, or pediatric endocrinologist locally. This requires a good transfer of all necessary information.

### 3.6. Patient Recruitment

The patient recruitment started in summer 2020. Thirty children/adolescents and their parents started the information management according to the developed materials. Ten families are still in the procedure of completing the checklist and several interviews for qualitative evaluation are outstanding.

## 4. Discussion

The Empower-DSD information management provides a new standardized information and care program for the first weeks after a suspected DSD diagnosis for children with genital variation and their parents. The structured care process is made transparent to all involved stakeholders via two tools in the specialized DSD center: guidelines and checklist for the multi-professional team and a personal health record for the families (“My record”). The link to primary care is endorsed with a third tool: a booklet for professionals outside a specialized DSD center.

The developed materials disclose relevant information to each target group and thereby support the family’s navigation through the medical system. To our knowledge, this is the first standardized program that equally addresses families and professionals.

The variability of DSD conditions—even with the same genotype—makes the use of a clinical algorithm difficult. Parents differ in their needs and ethical/social understanding of gender. Each family has different concepts of what is normal and how to deal with sexuality [6]. The requirements of children change with age [43]. To address this complexity in care, an individualized approach is necessary. Nevertheless, generic elements that recur in the care and education of families with a new DSD diagnosis were identified and sorted in a checklist. The medicalization of DSD is a point of criticism [44]. Psychosocial information is often perceived as missing by parents [45]. The checklist therefore equally values medical and psychosocial aspects of the care process.

The target groups’ varying demands were a challenge in the materials’ development. While specialists tend to wish for detailed information, parents wish for time to get used to the diagnosis and for practical information regarding daily life [46,47]. To address the different viewpoints of clinicians and individuals with DSD as well as their surrogates, patient support groups were actively included in the developmental process.

Several decision aids for families with newly diagnosed DSD have recently been developed. They focus on the decision-making of parents with newborn children and genital or gonadal surgery [48,49,50,51]. Three decision aids consist of checklists for the specialized DSD team. They include lists of topics to discuss with the family. One group developed a complex online tool for the families to navigate at home. Interestingly, this tool was only used by 40% of the families past the first log-in [49]. As one reason, the authors suspected technical problems but also different needs for the depth of information in different families. The information management of Empower-DSD aims to promote informed decision-making by providing education on decision-making and DSD. However, it cannot be seen as a decision-making aid, but as a toolkit to promote information exchange and standardized care. In doing so, it acknowledges that final decisions may or may not be made within the first few weeks.

German legislation banned cosmetic genital surgery on children with a DSD diagnosis in March 2021 and thereby changed the available options for families [52]. Surgery is still possible in life-threatening situations or in special cases after the authorization of a family court (e.g., sinus urogenitalis in CAH females). In all other cases, the decision for surgery must be deferred until an age where the child can participate in decision-making processes.

The information management’s contents were split into three separate tools. The latter were meant to provide only relevant information to each target group and avoid information overload. The structure of the information management is flexible and can be adapted to different DSD diagnoses, age groups, and clinic settings. All activities of the multidisciplinary team are documented, which is supposed to support the transition from pediatric to adult health care and thereby improve long-term health. A standardized documentation might also facilitate data entry into registries, as is increasingly demanded concerning rare diseases [24].

The developed materials are designed to be used in an interactive matter at the DSD clinic. The team can review information together with the family and tailor it to the patient’s and family’s needs. Depending on the family’s health literacy, information can be limited to basics or further resources can be shared.

The written information in “My record” complements face-to-face contact. Conversations with the team should extend the information and adapt it to the family’s cultural and educational background. The guideline to the DSD team is directed at professionals with experience in communication with families and children with DSD conditions. Research shows that the health provider’s communication skills affect parents’ decision-making processes [53]. These skills vary wildly, even amongst DSD specialists [54]. While the developed tool for professionals outside the DSD center encompasses tips on how to communicate with the family, no advice on this matter was included in the material for the multi-professional team in the DSD center. Details regarding communication for DSD specialists could be added to the information management in the future.

An effort has been made to integrate existing materials from German support groups into the information management. US and English patient support groups have created resources with reassuring information, such as dsdfamilies.org and the Handbook for Parents (Appendix A), which were not included because of the language barrier for the families. The previous project dsd-LIFE translated one brochure designed by dsdfamilies.org into German. Further collaboration in this style could improve the information of families with a new DSD diagnosis. On the other hand, cultural background, health systems, and national judicial regulations differ between the countries, which might impede the adaptation of international materials.

The approach to diagnostics and care in DSD is changing continuously. Ongoing research is to be expected [55]. Information quickly becomes outdated. As mentioned above, the German legislation regarding genital surgery changed since the information management´s first draft. Patient support groups reorganized and updated brochures, accordingly. Hence, materials and tools must be updated on a regular basis. This might be a barrier to the implementation of the developed tools in routine care. On the other hand, “My record” and the checklist are flexible. They can easily be adapted to changing guidelines or local requirements.

## 5. Conclusions

The newly developed information management for children with a recent DSD diagnosis standardizes the diagnostic and information process for patients, their families and professionals in three paper-based materials. It adapts the information to the families’ needs and to the health staff with different degrees of specialization. The materials navigate families through the medical system and endorse the documentation of the care process.

The information management’s evaluation will provide deeper insight into the needs and wishes of families and professionals during the first weeks of care.

## Figures and Tables

**Figure 1 jcm-11-03859-f001:**
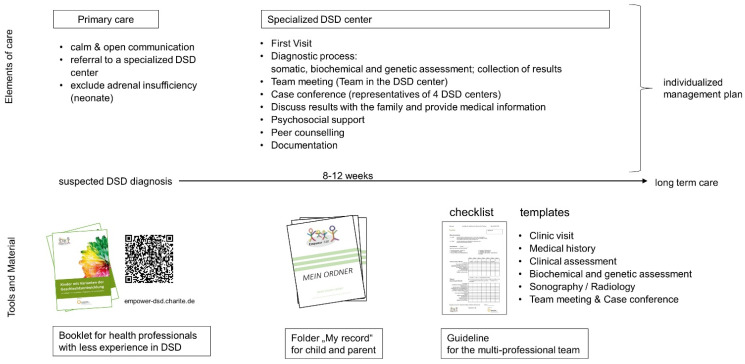
Elements of care and developed tools of the information management.

**Table 1 jcm-11-03859-t001:** Content of the booklet for health care professionals outside the specialized DSD center.

Medical background: What does the acronym DSD stand for? Why is the referral to a specialized center important? How quickly should a referral happen?—Urgent diagnostic procedures are only important to exclude CAH and other forms of adrenal insufficiency in the newborn. Guide for the communication with parents of newborn and older children/adolescents including useful phrases and wording. Contact to specialized centers and newborn screening labs in Germany. Contact to patient support groups and a list of existing patient information brochures.

**Table 2 jcm-11-03859-t002:** Checklist for the Empower-DSD information management.

**First Visit**			Date: _______________	
*Check with x if completed, otherwise leave blank*				
o explain goals of information management;			o introduce team members;	o concept of psychosocial care;
o concept of peer counselling;			o hand out “My record”;	o outline diagnostic process
Date	Date	Date		

**Diagnostic process**				
*x—completed; o—not completed; xx—previously performed; n—not applicable*				
			Medical history	
			Somatic assessment	
			Biochemical and genetic assessment	
			Sonography/imaging	
			Referral gynecology	
			Referral urology	
			Referral pediatric surgery	
**Diagnosis**				
*c—confirmed; u—unclear; s—suspected*				

**Discuss results; medical information**				
*x—topic discussed; o—not discussed; n—not applicable*				
			Basics: Sex development, gonads, steroid hormones	
			Urgency or non-urgency of treatment	
			Development of gender identity	
			Expected somatic development	
			Options for hormone treatment in puberty	
			Individual decision-making	
			Fertility	
			Malignant potential	
			Sexuality	
			Risks and benefits of surgery	
			Management plan	
**Psychosocial care parents**				
*x—topic discussed; o—not discussed; n—not applicable*				
			Address questions, fears, and concerns	
			Adopt a positive perspective on the child´s future	
			Each family has a different way to approach DSD.	
			Recommend the documentation of the decision-making process	
			View of biological sex and gender as continua	
			Diversity of physical appearance	
			Sex assignment	
			Communication within the social network (education and disclosure)	
			Communication with the healthcare system	
			Explaining DSD to the child in an age-appropriate way	
**Psychosocial care child**				
*x—topic discussed; o—not discussed; n—not applicable*				
			Address questions, fears and concerns	
			Adopt a positive perspective on the future	
			Education about the child´s body and condition	
			Diversity of physical appearance	
			Communication within the social network (education and disclosure)	
			Explain procedures in the specialized center	
			Explain care and guidance over time	
			Social and legal issues	
			Peer counselling	
**Team meeting/Case conference** (Professions of the attending participants and centers present are documented.)				
*x—topic discussed; o—not discussed; n—not applicable*				
			Results of assessments	
			Urgency of treatment	
			Evidence for recommendations	
			Management plan	
**Feedback to referring primary care provider**			Date: _________________	

**Table 3 jcm-11-03859-t003:** Sections and content of “My record”.

**Sections and Content**	**Purpose**
**Introduction**: Welcome letter to the family (different version for parents of a newborn and families with an older child).	
**Contact/Appointments**: Address, names of the multidisciplinary team members, space to note appointments	Personal Health record
**Visit in the DSD clinic**: What to expect during the visits at the DSD clinic. Template for taking notes during the visit or preparing questions for the next visit (for professionals and/or patients). What is a team meeting or a case conference? Role of photographs in the somatic assessment.	
**Results**: Short description of common diagnostic procedures in the DSD clinic and encouragement of the family to keep and file all results/discharge notes here.	
**Peer Counseling**: Introduction and contact to support groups, explanation of peer counseling.	Information
**Medical and psychosocial information**: Structured by age. Glossary of medical terms. Brochures of patient support groups designed for families with a recent diagnosis.	
**Thoughts and feelings**: Place for parents to write down thoughts, feelings, or information regarding the diagnosis process or DSD. This might be a diary entry or a letter to the child. For older children or adolescents, place for own thoughts or feelings.	Documentation of the decision-making process

## Data Availability

Not applicable.

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
