# Peer review of "Diagnosis of DSD in Children—Development of New Tools for a Structured Diagnostic and Information Management Program within the Empower-DSD Study"

_jcm, 2022, doi:10.3390/jcm11133859_

Round 1

Reviewer 1 Report

Dear Authors,

you presented an interesting and original paper. The idea of collecting written information and directing it to a) family, b) non specialized and c) specialized medical staff, is a very innovative concept. Four observations:

1) I don't find in "Results" section your preliminary results. In "Methods", pag. 4, lines 154-155, you write "the study aims for the inclusion of 30 children/adolescents and their parents". How many patients/family have you already analyzed? 

2) In "Methods" pag. 3, lines 97-98 you say that "One aim is to foster the diagnosis-specific knowledge and empowerment of patients and their parents, thereby promoting shared decision making". A shared therapeutic option with parents during neonatal period is not possible: according to German legislation, parents had to accept "wait and see" option if no urgent treatment are required. In this particular context there is no possibility of an shared decision-making approach, but only an informed approach. You well comment in "Discussion" pag. 9-10, lines 352-356. Please underline this aspect in "results" section too.

3) In "Methods" section, pag 7, lines 244-245 you write "Therefore, the guideline primarily defined the topics, not the team member's profession, who discuss the topic with the family".

Treating DSD patients is difficult, no unique algorithm is possible for therapy, often an individualized treatment is preferable. Why did you not standardize the team member assigned for discussing with family? Within the multidisciplinary team there are a lot of different competencies and that can differently affect parents/patients perception of the topic during discussion. Is always an expert of the same medical practice or is sometimes the "case manager" (pag. 7, line 242) for example a psychologist? Please explain that.

4) In "Results" section you mention Table S1 (pag 4, line 172 and pag 5 lines 208-209), I don't find the table in the manuscript.

Kind regards

Reviewer 2 Report

I recommend to explain more clearly at the beginning why it is necessary to develop such a model, although this has been added in the discussion. I also recommend adding a little more information about the project,  and a conclusion at the end of the text.
